# LEO: Learning Energy-based Models in Factor Graph Optimization

**Paloma Sodhi[1,2], Eric Dexheimer[1], Mustafa Mukadam[2], Stuart Anderson[2], Michael Kaess[1]**
[1]Carnegie Mellon University, [2] Facebook AI Research

**Abstract:**

We address the problem of learning observation models end-to-end for estimation. Robots operating in partially observable environments must infer latent states from multiple sensory inputs using observation models that capture the joint distribution between latent states and observations. This inference problem can be formulated as an objective over a graph that optimizes for the most likely sequence of states using all previous measurements. Prior work uses observation models that are either known a-priori or trained on surrogate losses independent of the graph optimizer. In this paper, we propose a method to directly optimize end-to-end tracking performance by learning observation models with the graph optimizer in the loop. This direct approach may appear, however, to require the inference algorithm to be fully differentiable, which many state-of-the-art graph optimizers are not. Our key insight is to instead formulate the problem as that of energy-based learning. We propose a novel approach, LEO, for learning observation models end-to-end with graph optimizers that may be non-differentiable. LEO alternates between sampling trajectories from the graph posterior and updating the model to match these samples to ground truth trajectories. We propose a way to generate such samples efficiently using incremental Gauss-Newton solvers. We compare LEO against baselines on datasets drawn from two distinct tasks: navigation and real-world planar pushing. We show that LEO is able to learn complex observation models with lower errors and fewer samples.

**Keywords:** factor graphs, energy-based learning, observation models

## 1  Introduction

We focus on the problem of learning observation models end-to-end for estimation. Consider a robot hand manipulating an object with only tactile feedback. It must reason over *a sequence of* touch observations over time to collapse uncertainty about the latent object pose. A common way to solve this is as an inference over a factor graph which relies on observation models that can map between states and observations [1, 2, 3]. However, in many domains, sensors that produce observations are complex and difficult to model. *Can we instead learn observation models from data?*

Given a batch of ground truth trajectories and corresponding measurements, how should we learn observations models? One approach would be to learn a direct mapping from measurement to state, for example as a regression or classification problem [4, 5, 6, 7]. However, while easy to optimize given a direct supervised loss, this only minimizes a surrogate loss independent of the graph optimizer and is not guaranteed to minimize the final tracking errors that we care about. The other option would be to directly minimize final tracking errors between optimized and ground truth trajectories [8, 9]. However, many state-of-the-art factor graph optimizers, e.g. iSAM2 [10], are not natively differentiable due to operations such as dynamic re-linearizations[1]. In such cases, we are limited to black-box search for learning parameters which is very sample inefficient [11, 12, 13].

Instead of differentiating through the optimization process, we note that what we ultimately care about is the final solution from the optimizer, which depends *only on the shape* of the optimized cost function. We would like a cost function that has low cost around the observed ground truth trajectories and high cost elsewhere. This is precisely what energy-based models aim to do by

---

Code and supplementary material can be found on https://psodhi.github.io/leo

5th Conference on Robot Learning (CoRL 2021), London, UK.

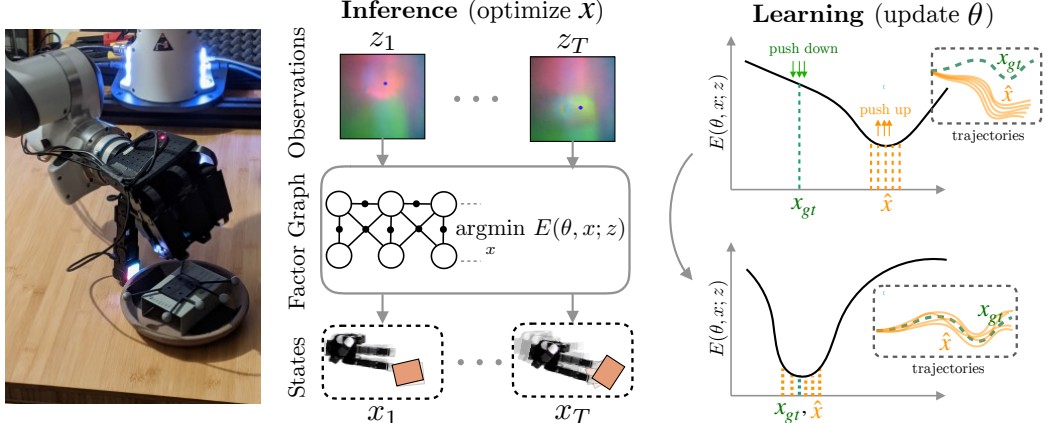

**Figure 1:** We show that learning observation models can be viewed as shaping energy functions that graph optimizers, even non-differentiable ones, optimize. **Inference** solves for most likely states $x$ given model and input measurements $z$. **Learning** updates observation model parameters $\theta$ from training data.

shaping an "energy" function to be low around observed data and high elsewhere [14, 15, 16]. We cast our problem of learning observation models for estimation as energy-based learning.

We propose a novel approach, **L**earning **E**nergy-based Models in Factor Graph **O**ptimization (LEO), for learning observation models end-to-end with graph optimizers that may be non-differentiable. LEO alternates between sampling trajectories from the graph posterior and updating the model to match the distribution of samples to ground truth trajectories. We show that we can generate such samples efficiently using incremental Gauss-Newton graph solvers that create a Gaussian approximation for the Boltzmann distribution defined over a continuous space of trajectories. To the best of our knowledge, this is the first paper to show that even non-differentiable factor graph models can be learned efficiently. Our main contributions are:

1. A novel formulation of learning observation models in estimation as energy-based learning.

2. An algorithm, LEO, to learn observation models that minimize end-to-end tracking errors in factor graph optimizers that may be non-differentiable.

3. Empirical evaluation on two distinct tasks: synthetic navigation and real-world planar pushing.

## 2   Related Work

**Filtering and smoothing based estimation.** Observation models represent joint or conditional distributions between states and measurements. Early estimation methods utilized these models within filtering contexts such as Kalman filters and EKFs/UKFs [17, 18]. Recent methods have also looked at making these filters differentiable for end-to-end learning [19, 20, 21, 22, 23]. However, filtering can be inconsistent for nonlinear estimation problems due to linearization based on past, marginalized states that cannot be undone [24]. Instead, a wide consensus amongst the simultaneous localization and mapping (SLAM) community is to solve these problems as smoothing or nonlinear optimization objectives [3, 2]. Unlike filtering, smoothing allows dynamic relinearization of past states as well as exploits the inherent sparsity resulting in efficient and more accurate solutions. Factor graphs are an increasingly popular way for solving such smoothing problems [1, 2, 25, 4, 5, 26, 27, 28, 29], offering a flexible modeling framework while being efficient to optimize. We use the factor graph based smoothing framework in this paper.

**Learning for smoothing.** Typically, observation models used in factor graphs are analytic models defined a-priori [30, 31, 32]. Recent work has looked at using learned models [4, 5, 6] or learned measurement representations [26, 33] either to be used independently or to be plugged into a graph optimizer. However, these model parameters are learned on surrogate losses independent of the graph optimizer, and hence do not directly attempt to minimize final tracking errors.

An alternative approach is to view observation model parameters as hyper-parameters of the graph and perform a hyper-parameter search. This is usually done as sampling and derivative-free ap-

---

[1]See Appendix A for discussion on applying LEO to differentiable optimizers.

proaches using black-box solvers [11, 12, 13]. However, the high computational burden associated with sampling and evaluation can be prohibitive for large parameter spaces.

**Differentiable optimization.** Instead of a search, one may learn these model parameters via a continuous gradient-based optimizer matching graph optimizer output against a desired solution. Minimizing this loss would, however, require differentiating through the argmin optimization process. One way to do this is via *unrolling* where the optimization is represented as a series of differentiable, gradient based updates [16, 34, 35]. Since smoothing is a nonlinear least-squares optimization, it too can be unrolled in a similar fashion via differentiable Levenberg-Marquardt updates [9, 36, 37] or differentiable Gauss-Newton updates [8, 38, 39]. However, vanilla unrolling suffers from some drawbacks, notably the learned cost function can be sensitive to the specific optimization procedure, e.g. number of unrolling steps [40]. Moreover, this approach cannot be readily applied to commonly used optimizers relying on non-differentiable heuristics such as those in GTSAM, SNOPT [41, 42]. An alternate to unrolling is to employ the implicit function theorem which states that an optimal energy function must satisfy first-order necessary conditions [43, 44]. While this is promising for convex problems, for non-convex problems, one may recover an energy function with multiple minima such that the optimizer at test time may find itself in a different basin from ground truth.

**Energy-based learning.** Instead of differentiating through the optimization process, we note what we ultimately care about is the final solution from the optimizer, which depends only on the shape of the cost function. This is what energy-based models aim to do by shaping an "energy" function to be low around observed data and high elsewhere [14]. Energy-based modeling is a family of unsupervised learning methods popular in many structured prediction tasks such as inverse reinforcement learning [45, 46] and computer vision [47, 48]. Recently, a number of approaches have looked at training deep energy-based models [49, 50, 16]. The challenge with such methods is generating samples from the energy based model. Methods either rely on MCMC sampling [51, 52, 53] which does not scale well with dimension or resort to learning a separate generator [54, 55, 15]. We propose a way to generate such samples efficiently using incremental Gauss-Newton solvers.

## 3 Problem Formulation

We formalize our problem of learning observation models under the general theoretical framework of energy based models. We begin by describing the observation model, how it is optimized at inference time and finally how its parameters are learned from training data (Fig. 1).

### 3.1 Model

We define an energy-based model that captures dependencies between variables in a graph by associating a scalar energy $E(\cdot)$ to each configuration of the graph. Our goal is to model the likelihood of a sequence of latent states $x$ given a sequence of measurements $z$. We adopt a factor graph framework (Fig. 1(a)) for expressing this likelihood as[2],

$$P_\theta(x|z) = \frac{1}{\mathcal{Z}(\theta;z)} \exp\left\{-E(\theta,x;z)\right\} = \frac{1}{\mathcal{Z}(\theta;z)} \exp\left\{-\frac{1}{2}\sum_k ||f(\theta,x_k;z_k)||^2\right\} \quad (1)$$

where, $x := \{x_1 \ldots x_T\}$ is the sequence of latent states, and $z := \{z_1 \ldots z_T\}$ is the sequence of measurements. $||f(\theta,x_k;z_k)||^2$ are the factor costs with $f(\theta,x_k;z_k)$ being the local observation model mapping a *subset* of states $x_k \subseteq x$ and measurements $z_k \subseteq z$ to a cost, $\theta$ are the learnable model parameters, and $\mathcal{Z}(\theta;z) = \int_x \exp(-E(\theta,x;z))dx$ is the normalization or partition function.

Note that Eq. 1 is a special case of the Boltzmann distribution where the energy $E(\cdot)$ is a sum-of-squares. Notably, when $f(\theta,x_k;z_k)$ is linearized, $E(\cdot)$ is quadratic in $x$ and subsequently $P_\theta(x|z)$ becomes a Gaussian distribution. This is the structure that Gauss-Newton solvers employ, and we will leverage this fact to efficiently sample trajectories as described later in Section 4.3.

### 3.2 Inference

At inference time, given a sequence of measurements $z$, we wish to solve for the most likely sequence of latent states $x$. We formulate this as maximizing the log-likelihood of the model,

$$\hat{x} = \underset{x}{\operatorname{argmax}} \, \log P_\theta(x|z) \quad (2)$$

---

[2]We use the word likelihood as in typical learning formulations. Specifically, it refers to the posterior of states given measurements. Model parameters $\theta$ include covariances $\Sigma_k$ in the norm $||\cdot||_{\Sigma_k}$ as well.

Substituting the model expression from Eq. 1, and dropping constants $\mathcal{Z}(\theta; z)$ and $1/2$, results in a nonlinear least-squares minimization of the form,

$$\hat{x} = \underset{x}{\operatorname{argmax}} \, \log \frac{1}{\mathcal{Z}(\theta; z)} \exp \left\{ -\frac{1}{2} \sum_k ||f(\theta, x_k; z_k)||^2 \right\} = \underset{x}{\operatorname{argmin}} \sum_k ||f(\theta, x_k; z_k)||^2 \qquad (3)$$

We use an efficient online Gauss-Newton graph solver for this objective. This enables both real-time inference as well as efficient sampling for learning model parameters during training (Section 4.3).

### 3.3 Learning

At train time, our goal is to learn a model that explains training data of pairs $(x_{gt}, z)$. We express this as minimizing a loss $\mathcal{L}(\theta)$ over the training data

$$\mathcal{L}(\theta) = \frac{1}{|\mathcal{D}|} \sum_{(x_{gt}^i, z^i) \in \mathcal{D}} \mathcal{L}(\theta; x_{gt}^i, z^i) \qquad (4)$$

where, $\{x_{gt}^i, z^i\} \in \mathcal{D}$ is a training dataset of ground truth trajectories and measurements.

There are various choices of losses $\mathcal{L}(\cdot)$ for learning model parameters $\theta$,

**Loss 1: Minimize final tracking loss.** A straightforward choice for $\mathcal{L}(\cdot)$ is to directly minimize final tracking errors between optimized $\hat{x}^i$ and ground truth $x_{gt}^i$ trajectories, i.e. $\mathcal{L}(\theta; x_{gt}, z) = 1/|\mathcal{D}| \sum_i ||\hat{x}^i \ominus x_{gt}^i||_2^2$. While this loss directly minimizes final tracking errors, it may not be differentiable since the graph inference path from $\theta$ to $\hat{x}$ is not differentiable. In such cases, we are limited to black-box hyperparameter search which is very sample inefficient.

**Loss 2: Minimize surrogate loss.** An alternate choice for the loss $\mathcal{L}(\cdot)$ is to directly map measurements to ground truth states, for example as a regression or classification problem. Such a surrogate loss is independent of graph inference, and unlike Loss 1, is differentiable and easy to optimize. However, this does not minimize the final tracking errors that we care about.

**Loss 3: Minimize energy-based loss.** Finally, we consider a loss that is a function of the energy, i.e. $\mathcal{L}(E(\theta, \cdot); x_{gt}^i, z^i)$. Intuitively, this assigns a low loss to *well-behaved* energy functions, i.e. functions that give the lowest energy to training data of ground truth trajectories (correct answers) and higher energy to unseen data (incorrect answers).

This energy-based loss, unlike Loss 2, is highly correlated with the final tracking loss as it shapes the energy (or cost) landscape so as to make the inference step return trajectories closer to the ground truth. Moreover, unlike Loss 1, this loss is only a function of energies which is differentiable *irrespective of inference*. Hence, we use Loss 3 and detail our approach in the next section.

## 4 Approach

We instantiate the broad framework of energy-based learning within the context of factor graph estimation in 4.1 and 4.2. In 4.3 we show how incremental Gauss Newton approaches enable efficient sampling from these energy based models.

### 4.1 Normalized Negative Log-Likelihood Loss

We define the overall loss as the negative log-likelihood (NLL) of the data under the energy model[3]:

$$\mathcal{L}(\theta) = \frac{1}{|\mathcal{D}|} \sum_{(x_{gt}^i, z^i) \in \mathcal{D}} -\log P_\theta(x_{gt}^i | z^i) \qquad (5)$$

Substituting $P_\theta(x|z)$ as the Boltzman distribution expression from Eq. 1 we have,

$$\mathcal{L}(\theta) = \frac{1}{|\mathcal{D}|} \sum_{(x_{gt}^i, z^i) \in \mathcal{D}} E(\theta; x_{gt}^i, z^i) + \log \int_x \exp(-E(\theta; x, z^i)) dx \qquad (6)$$

---

[3]The motivation for NLL loss comes from probabilistic modeling, particularly maximum entropy moment matching. We provide a more detailed derivation along with its Gaussian approximation in Appendix C.

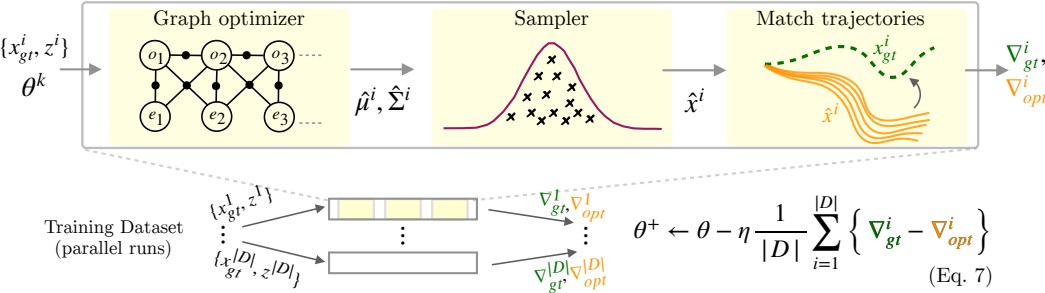

**Figure 2:** LEO algorithm: For each training example, the zoomed in panel shows the three main stages: (a) solve graph optimization, (b) sample trajectories from graph posterior, and (c) compute gradients. Gradients are then pooled together to update parameters $\theta$.

---

**Algorithm 1** LEO algorithm

---

**Input** Initial $\theta$, Training data $\{x_{gt}^i, z^i\}_{i=1}^{|\mathcal{D}|}$
**while** `<!converged>` **do**
    **for** each training example i **do**
        $\hat{\mu}^i, \hat{\Sigma}^i = \underset{x}{\operatorname{argmin}}\{E(\theta, x; z^i)\}$                                                   ▷ Solve graph optimizer objective
        $\hat{x}^i \sim \mathcal{N}(\hat{\mu}^i, \hat{\Sigma}^i)$                                                         ▷ Sample $S$ trajectories from graph posterior
    **end for**
    $\theta^+ \leftarrow \theta - \eta \frac{1}{|\mathcal{D}|} \sum_{i=1}^{|\mathcal{D}|} \left\{ \nabla_\theta E(\theta; x_{gt}^i, z^i) - \frac{1}{S} \sum_{\hat{x}^i} \nabla_\theta E(\theta; \hat{x}^i, z^i) \right\}$   ▷ Update model parameters (Eq. 7)
**end while**

---

The expression above has an intuitive explanation. The first term penalizes high energy over ground truth training samples. The second term penalizes low energy everywhere else, thus acting as a "contrastive" term. This has a desirable effect of globally shaping the energy function so that the minima lies at the training samples. The contrastive term prevents the energy surface from becoming flat and thus degenerate [14]. Unfortunately, we cannot directly plug in Eq. 6 as a standard machine learning loss (e.g. in PyTorch) to optimize. This is because the second log partition term is over a continuous space of trajectories $x$, which renders exact computation intractable for arbitrary nonlinear cost functions [56].

Instead, let us take the gradient of this loss, and substitute $P_\theta(x|z^i)$ expression from Eq. 1,

$$\nabla_\theta \mathcal{L}(\theta) = \frac{1}{|\mathcal{D}|} \sum_{(x_{gt}^i, z^i) \in \mathcal{D}} \left[ \underbrace{\nabla_\theta E(\theta; x_{gt}^i, z^i))}_{\text{ground truth samples}} - \underset{\substack{x \sim \\ P_\theta(x|z)}}{\mathbb{E}} \underbrace{\nabla_\theta E(\theta; x, z^i))}_{\text{learned distribution samples}} \right] \tag{7}$$

We see the log partition term transforms into an expectation over the learned distribution. The gradient expression above has a similar interpretation as the loss. The first term pushes down the energy of ground truth training samples. Interestingly, the second term now pulls up the energy of samples drawn from the learned Boltzmann distribution. If one is able to generate such samples, computing the gradient is straightforward.

Generally, sampling from the continuous Boltzmann distribution is intractable, relying typically on MCMC methods. But as we will detail in Section 4.3, we can indeed generate such samples efficiently since our graph optimizer maintains a Gaussian approximation of this distribution. Additionally in practice, we found that introducing a temperature term $T$ inside the Boltzmann distribution, i.e. $P_\theta(x|z) \propto \exp(-1/T E(\theta, x; z))$, was critical for controlling the variance of the samples. Setting $T = 0$ corresponds to only using the mode of the distribution which in this case is the mean trajectory. This corresponds to minimizing a generalized perceptron loss [14].

## 4.2 LEO Algorithm

Algorithm 1 summarizes the LEO algorithm, also illustrated in Fig. 2. It takes as inputs the training data containing ground truth state and measurement pairs, i.e. $\{x_{gt}^i, z^i\}_{i=1}^{|\mathcal{D}|}$, along with initial values for the learnable observation model parameters $\theta$. The goal is to converge to a $\theta$ that minimizes the log-likelihood loss in Eq. 5 over the given training set $i = 1 \dots |\mathcal{D}|$.

LEO invokes a black-box, non-differentiable graph optimizer iSAM2 in the loop to generate samples. Within each LEO iteration, for every training example $i$, we invoke the graph optimizer to obtain a Gaussian approximation $\mathcal{N}(\hat{\mu}^i, \hat{\Sigma}^i)$ to the original Boltzmann distribution $P_\theta(x|z)$ about its mode. We then sample trajectories $\hat{x}^i \sim \mathcal{N}(\hat{\mu}^i, \hat{\Sigma}^i)$ from this distribution.

The next step is to compute gradients of the energy, i.e. $\nabla_\theta E(\theta; x_{gt}^i, z^i)$ and $\nabla_\theta E(\theta; \hat{x}^i, z^i)$, at the ground truth $x_{gt}^i$ and the samples $\hat{x}^i$, respectively. Note that the gradients here are only w.r.t. $\theta$, and not $x$, and hence do not require the graph optimizer to be differentiable. Once we have the two gradients, we use the difference between them to update the observation model parameters $\theta$. These updated parameters are then used in the next LEO iteration.

## 4.3   Efficient Inference via Incremental Gauss-Newton

The only thing that remains to be specified in Algorithm 1 is — How do we generate samples efficiently from the graph inference? In Section 3.2 we noted that our inference algorithm must satisfy two requirements (a) real-time inference and (b) sampling from the Boltzmann distribution. We achieve this using via incremental Gauss-Newton to solve the nonlinear least squares in Eq. 3.

Gauss-Newton proceeds by linearizing observation model functions $f(\cdot)$ about a point $x_k^0$,

$$f(\theta, x_k; z_k) = f(\theta, x_k^0 + \delta x_k; z_k) \approx f(\theta, x_k^0; z_k) + F_k \delta x_k \tag{8}$$

where, $x_k{}^0$ is the linearization point, $F_k = \left. \frac{\partial f(\theta, x_k; z_k)}{\partial x_k} \right|_{x_k^0}$ is the measurement Jacobian, and $\delta x_k = x_k - x_k^0$ the state update vector. Measurements $z$ and learnable parameters $\theta$ are treated as constant within the graph optimization step. Substituting the Taylor expansion in Eq. 8 back into Eq. 3,

$$\delta x^* = \operatorname*{argmin}_{\delta x} \sum_k ||F_k \delta x_k - \left( -f(\theta, x_k^0; z_k) \right)||^2 = \operatorname*{argmin}_{\delta x} ||A \delta x - b||_2^2 \tag{9}$$

where, $A$, $b$ are constructed by concatenating all $F_k$, $f(\theta, x_k^0; z_k)$ together into a single matrix and vector respectively. We then iterate $x^0 \leftarrow x^0 + \delta x^*$ until convergence.

We can solve for the objective in Eq. 9 efficiently by exploiting two properties of the problem: *sparsity* and *online* measurements. For SLAM problems, both $A$ and $A^T A$ are large but sparse matrices, and can hence be factorized efficiently using sparse QR or Cholesky factorization respectively [57]. However, this still does not guarantee real-time inference during test time since $A$ grows with measurements over time, making the method increasingly slower. Instead, subsequent incremental solvers [10, 58, 59] reuse matrix factorizations from previous time steps. In this work, we use iSAM2 [10], an incremental Gauss-Newton graph solver whose resulting MAP solution can be expressed as a Gaussian probability distribution with mean and covariance,

$$\hat{\mu} = \operatorname*{argmin}_x \sum_k ||f(\theta, x_k; z_k)||^2, \qquad \hat{\Sigma} = \left( A^T A \right)^{-1} |_{x=\hat{\mu}} \tag{10}$$

This is the Gaussian approximation of the original Boltzmann distribution $P_\theta(x|z^i)$ which we can now sample from efficiently. The graph posterior covariance $\hat{\Sigma}$ for such a Gauss-Newton solver is obtained using just Jacobians $A$ without needing to compute a Hessian [2]. Note that sampling $\hat{x} \sim \mathcal{N}(\hat{\mu}, \hat{\Sigma})$ is done at the tangent space of the linearized graph, i.e. $\hat{x} \leftarrow \hat{\mu} \oplus \delta x$ where $\delta x \sim \mathcal{N}(0, \hat{\Sigma})$ and $\oplus$ is a retraction [60]. The Gauss-Newton method for inference is an efficient technique for sampling from intractable models. A recent overview of the Laplace approximation with Gauss-Newton method can be found in [61]. Here, we build a specific Laplace approximation using the incremental solver iSAM2 that scales to sampling efficiently in real-time from very large graphs.[4]

## 5   Results and Evaluation

We evaluate LEO on synthetic navigation and real-world planar pushing. We compare against baselines on metrics like final tracking error and sample efficiency. We implement LEO in PyTorch [62] and interface it with the GTSAM C++ library [41], which contains several non-differentiable factor graph optimizers. We use the iSAM2 [10] optimizer for real-time, online optimization.

---

[4]See Appendix B for run time comparison to other samplers.

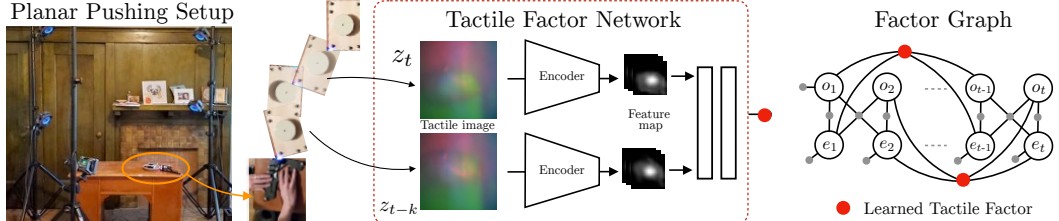

**Figure 3: (a)** Experimental setup for real-world planar pushing, **(b)** Learned tactile factor (red) penalizes deviations between predicted relative pose from the tactile factor network and estimated relative pose using current variable estimates in the graph.

## 5.1 Overview

*Experimental setup:* For synthetic navigation (Datasets N1–N4), the goal is to estimate latent robot poses $x_t \in SE(2)$ from relative odometry and absolute GPS measurements. For real-world planar pushing (Datasets P1–P3), the goal is to estimate latent object poses $o_t \in SE(2)$ from tactile image measurements using a DIGIT tactile sensor [63] as shown in the setup in Fig. 3. Both these problems are modeled as factor graphs with poses as variable nodes and measurements as factor nodes.

*Observation model parameters:* Recall factor costs are defined by the local observation model $f(\cdot)$ and learned model parameters $\theta$. Since we use Gaussian factors, we will learn their mean and covariance parameters, $\theta := \{\mu, \Sigma\}$. For the navigation task, we learn (a) fixed co-variances $\theta := \{\Sigma_{odom}, \Sigma_{gps}\}$ (N1, N2), and (b) covariances as a function of measurements $\theta := \{\Sigma_{odom}(z), \Sigma_{gps}(z)\}$ (N3, N4). For the planar pushing task, we learn (a) fixed covariances $\theta := \{\Sigma_{tac}, \Sigma_{qs}\}$ (P1, P2), and (b) means as a function of tactile image measurements $\theta := \{\mu_{tac}(z)\}$ (P3). For all experiments, we use a diagonal covariance model.

*Baselines:* We compare against a learned sequence model such as LSTM, and black-box hyperparameter search methods such as CMAES [11] and Nelder-Mead, all of which optimize Loss 1. We also compare against a surrogate supervised learning method [4] which optimizes Loss 2.

Appendix D provide more details on the factor graph models, setup, and additional results.

## 5.2 End-to-end Tracking Errors

Fig. 4 shows final tracking results for navigation datasets. We initialize LEO with random $\theta$ such that the graph optimizer returns trajectories far from the ground truth. At the final iteration, LEO converges to a $\theta$ that matches the optimized trajectories to ground truth. We summarize quantitative performance against baselines minimizing final tracking loss (Loss 1). LEO consistently outperforms baselines on final rmse tracking errors. The difference is more pronounced for datasets N3, N4 where the varying covariance model has a higher dimensional parameter space.

Fig. 5 shows final tracking results for real-world planar pushing datasets. Similar to navigation, we show LEO converging to ground truth trajectory for two different objects, a rectangle and ellipse, with differing local contact patches. We report quantitative performance on datasets P1, P2. LEO outperforms different hand-tuned choices of covariances as used in prior work [4].

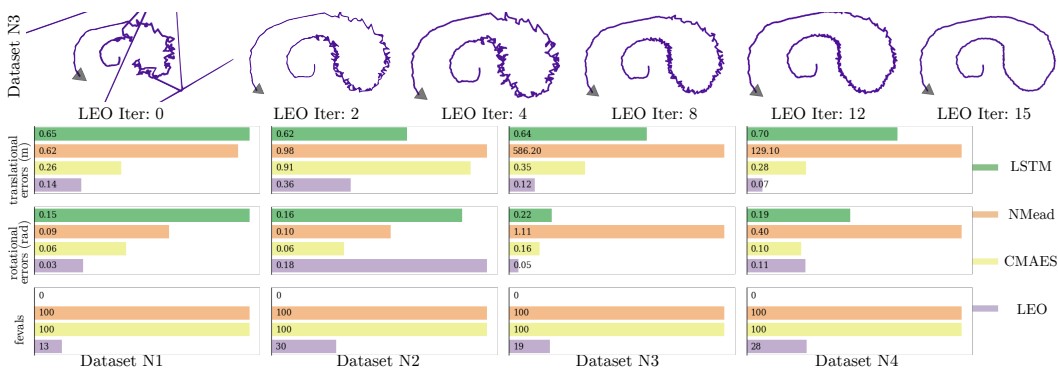

**Figure 4: Synthetic Navigation Datasets**: Comparison of LEO against baselines. Each dataset N1-N4 consists of 50 varying trajectories with a 30/20 train/test split. Translational and rotational trajectory tracking errors represent average RMSE over the test set. fevals is number of black-box graph optimizer calls and is computed per data point during training.

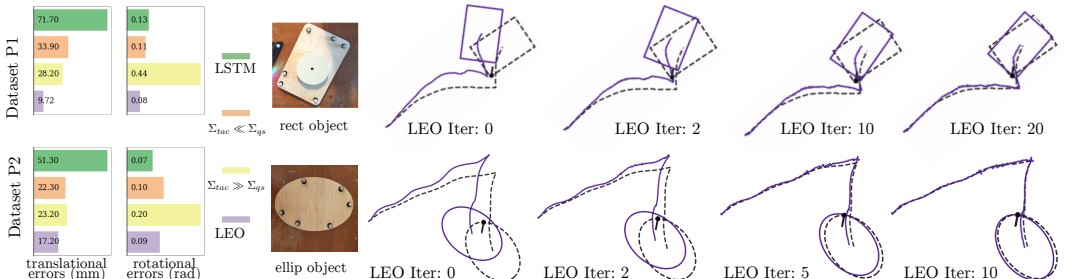

**Figure 5: Real-world Planar Pushing Datasets**: Each dataset P1, P2 consists of 15 varying trajectories with a 10/5 train/test split for rect, ellipse object respectively. Translational and rotational trajectory tracking errors represent average RMSE over the test set.

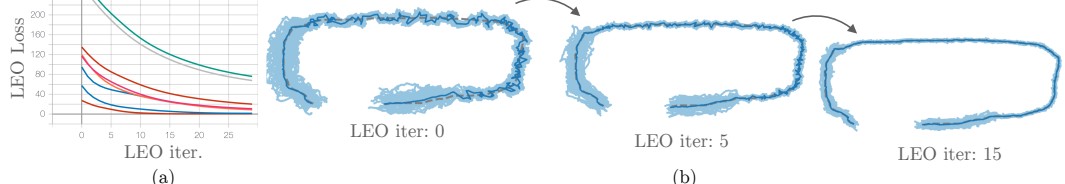

**Figure 6: (a)** LEO convergence for varying initial $\theta$ on dataset N1. **(b)** Evolution of graph optimizer trajectory samples over LEO iterations.

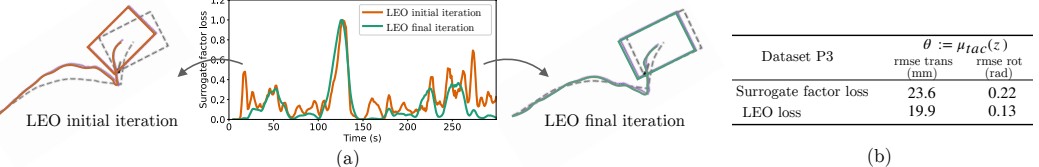

**Figure 7: (a)** Analysis showing that a surrogate loss, independent of the graph optimizer, may not minimize end-to-end tracking errors. **(b)** Minimizing only surrogate loss results in a higher final tracking error than minimizing the LEO loss.

## 5.3 Sample Efficiency and Convergence

Black-box hyper-parameter search is sample inefficient and does not scale well with increasing dimensionality. LEO, on the other hand, is efficient as it uses gradients. This can be seen in Fig. 4, where LEO consistently has lower graph optimizer calls (fevals). Moreover, LEO can also be used to learn models with large parameter spaces, e.g. Fig. 7 learns parameters for a tactile factor neural network mapping image features to transformations.

Finally, Fig. 6 shows LEO consistently converging for varying $\theta$ initializations on navigation dataset N1. We further show how the LEO samples evolve over iterations. Initially, when the cost is incorrect, the samples are more spread out. Over iterations, as the cost converges, the samples concentrate about the ground truth trajectory.

## 5.4 Surrogate Loss Analysis

We further analyze why a surrogate loss (Loss 2), independent of the graph optimizer, may not minimize end-to-end tracking errors. We train a network on surrogate supervised learning loss to directly map a pair of tactile image features to ground truth poses similar to prior work [4]. Fig. 7(b) shows the per time step surrogate loss on two trajectories — an initial trajectory that deviates from ground truth, and a final converged trajectory. While the tracking errors of final is much less than initial, the surrogate loss does not strictly improve. This is further reflected in the table in Fig. 7(c) where the surrogate loss baseline has higher final tracking errors over LEO for dataset P1.

## 6 Conclusion

We presented an approach, LEO, for learning observation models end-to-end with graph optimizers that may be non-differentiable. We compared LEO against baselines for different observation model types on two distinct tasks of navigation and real-world planar pushing. Overall, LEO is able to learn observation models with lower errors and fewer samples. While we investigated LEO with non-differentiable optimizers in this paper, as future work, we would like to apply LEO to differentiable optimizers and compare against alternate solutions such as unrolling (see Appendix A for a more detailed discussion). We would also like to scale to more complex observation models with applications to 3D object tracking using tactile sensors.

## Acknowledgements

This work was supported through the Facebook FRAIM program. We thank Frank Dellaert for insightful feedback and suggestions on the paper.

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
