# OpenReview forum: "LEO: Learning Energy-based Models in Factor Graph Optimization"
_robot-learning.org/CoRL/2021/Conference — CoRL2021 Poster_

### Official Review · Reviewer_uKGu · 2021-07-04

**Originality:** Fair
**Technical Quality:** Good
**Clarity Of Presentation:** Very Good
**Impact:** 3

**Recommendation:**

Weak Reject: I recommend rejecting the paper, but will not argue for my recommendation if the majority of other reviewers have a different opinion.

**Summary:**

This paper applied energy-based method to solve a factor-graph optimization task (e.g., graphs that comprise factor nodes representing latent states which are connected to observation nodes). The proposed method has concrete robotic applications such as learning an observation model for a scenario concerning a robot hand manipulating an object with only tactile feedback.

The paper in particular focuses on two key points: (1) optimizing factor graph via implicit energy-based loss instead of explicit surrogate loss (i.e., using losses that parameterize the compatibility between latent states & observations instead of losses that characterizes the difference between ground-truth and explicit mapping from observations to latent states); and (2) treating factor graphs as random variables instead of fixed, unknown quantities, which relax the original discrete optimization task (over the discrete space of graphs) to a continuous optimization problem (over the continuous space of parameters defining the graph distribution).

**Issues:**

Please refer to my specific comments in the Weakness section above.

**Reviewer Expertise:**

Fair: Some knowledge of the area

**Strengths And Weaknesses:**

STRENGTH:

The paper is well-written &  the mathematical exposition is not difficult to follow. The key points of the paper are well-communicated.

The proposed method has also be extensively evaluated on several datasets with clear visual demonstrations, which are really helpful in highlighting the usefulness of the proposed method

WEAKNESSES:

While this is a nice application paper, it does not exactly provide any new algorithmic and/or theoretical insights.

In fact, both key points of (1) optimizing via energy-based implicit loss; and (2) optimizing distribution of graphs instead of graphs themselves to convert a discrete optimization task into a standard gradient-based optimization problem have been proposed and applied extensively in many fields of ML.

In addition, the comparison with optimizing surrogate explicit loss appears minimal and lacks specific detail. Other than the vague description in line 139, I couldn't find where else in the main text such explicit loss was detailed. Aren't there existing works on the specific applications used in the experiment that the authors should have compared with?

Last, I'd suggest improving the differentiation between Loss 2 & Loss 3. Claiming that Loss 2 is surrogate & Loss 3 is energy-based (hence, implying it is not a surrogate) seems a bit misleading -- to me, the key difference is Loss 2 is based on an explicit mapping from observations to states while Loss 3 only characterize a fitness between a state configuration & the observations, which is implicit regarding the relationship between observations & latent states.

**Summary Of Recommendation:**

I think the paper is technically correct, well-motivated for an application but the entire work appears very incremental and is unlikely to have much impact regarding algorithmic novelties (e.g., both key points of the paper have been investigated and applied heavily in the ML but perhaps with different context)

I have elaborated my position in the "weaknesses" section above. That being said, I am not an expert in robotic so I am not entirely sure that this paper should be judged purely from a ML perspective -- if the other reviewers feel that the practical impact of this paper is sufficient in the robotic context, I would be happy to upgrade my rating.

But, regardless of the final outcome, I would still urge the authors to further elaborate & expand their comparison benchmark (especially against discrete graph optimizer approaches & existing works that employ explicit losses).

---

> ### Author Response · Authors · 2021-08-26
> **Response to Reviewer uKGu**
>
> We thank the reviewer for their detailed and thoughtful feedback. We appreciate that the reviewer found the paper well-communicated and well-evaluated. We have tried to address the main concerns and recommendations below:
>
> **Novelty**
>
> We agree with the reviewer that energy-based models are a broad framework used across many unsupervised learning applications. However, to the best of our knowledge, this is the first paper to make a connection between learning observation models end-to-end in state estimation / SLAM and energy-based learning. While factor-graph frameworks are widely used in SLAM and state estimation [1,2], there is very little work in the literature for learning observation or sensor models in factor graphs end-to-end -- typically models are hand-designed or learned independently.
>
> The only such work learning models end-to-end require factor graph optimizer to be differentiable and do this via unrolling [3,4]. However, most real-time SOTA factor graph libraries are not differentiable. Moreover, our initial analysis in Appendix A shows that differentiable unrolling approaches can easily overfit on simple 1D problems and fail to recover the correct model.  On the other hand, since we only learn the shape of the energy function which is independent of the optimizer, our approach is robust to the optimization procedure.
>
> In the context of energy-based methods, our novelty lies in the specific way we build a Laplace approximation using a popular “incremental” Gauss-Newton solver iSAM2 [5] in SLAM that can scale to sampling efficiently from very large graphs. The key insight is to leverage sparsity and incrementally construct the Jacobian as new measurements arrive thereby enabling real-time solutions.
>
> Finally, in the context of real-world tactile planar pushing application, this is the first paper to argue and show that observation models can be learned end-to-end. The previous SOTA method [6] learned observation models independently and showed that it outperformed hand-designed baselines. We go a step further and show evaluations where learning observation models end-to-end outperforms this approach.
>
> **Discrete optimization over the graph structure**
>
> There are two aspects to the problem: What edges (factors) to add to the graph and the weight of each edge (factor) in the graph? We focus on learning the latter, i.e. the weight of each edge given the graph structure. For the former, we assume that the structure of the quasi-static motion and geometric factors for planar pushing is known from physics. The tactile factors are added over a window size determined heuristically. We have added more details on this graph structure in Appendix C. We have also updated the paper draft to include a reference [7] from the former and are happy to include any other references that the reviewer had in mind.
>
> **Details on Loss 2**
>
> We provide more details on Loss 2 in Appendix C. This loss has been used in prior work [6] that shows that learning with Loss 2 outperforms non-learning based baselines typically used for solving the planar pushing estimation problem. Hence, we only compare against this Loss 2.
>
> Loss 2 is indeed a surrogate loss because we are not reasoning about errors in the space of trajectories but instead looking at independent pairs of observations to states. For instance, other than the learned tactile factors in the graph, we also have other factors in the graph such as quasi-static motion and geometric intersection factors. Loss 2 does not reason about how to trade-off between these various factors. On the other hand, Loss 3 takes into account the impact of the particular observation model on the final optimized trajectory that optimizes all the factors. In this sense, Loss 3 is highly correlated with the final trajectory tracking errors compared to Loss 2.
>
> References
>
> [1] F. Dellaert and M. Kaess. Factor graphs for robot perception. Foundations and Trends in Robotics, 2017.
>
> [2]​​ C. Cadena, L. Carlone, H. Carrillo, Y. Latif, D. Scaramuzza, J. Neira, I. Reid, and J. J. Leonard.  Past, present, and future of simultaneous localization and mapping:  Toward the robust-perception age. IEEE Trans. Robotics, 2016.
>
> [3] B. Yi, M. Lee, A. Kloss, R. Martin-Martin, and J. Bohg. Differentiable factor graph optimization for learning smoothers. arXiv preprint arXiv:2105.08257, 2021.
>
> [4] M. Bhardwaj, B. Boots, and M. Mukadam. Differentiable Gaussian process motion planning.  In IEEE Intl. Conf. on Robotics and Automation (ICRA), 2020.
>
> [5] M. Kaess, H. Johannsson, R. Roberts, V. Ila, J. Leonard, and F. Dellaert. iSAM2: Incremental smoothing and mapping using the Bayes tree. Intl. J. of Robotics Research, 2012.
>
> [6] P.  Sodhi,  M.  Kaess,  M.  Mukadam, and  S.  Anderson. Learning tactile models for factor graph-based estimation. In IEEE Intl. Conf. on Robotics and Automation (ICRA), 2021
>
> [7] B. Wilder, E. Ewing, B. Dilkina, and M. Tambe. End-to-end learning and optimization on graphs. NeurIPS 2019.

---

### Official Review · Reviewer_bQ4N · 2021-07-23

**Originality:** Very Good
**Technical Quality:** Very Good
**Clarity Of Presentation:** Very Good
**Impact:** 4

**Recommendation:**

Weak Accept: I recommend accepting the paper, but will not argue for my recommendation if the majority of other reviewers have a different opinion.

**Summary:**

The paper presents a method for learning observation models, i.e. probabilistic representations of the relationship between a latent state and sensor data, based on combining energy models with incremental Gauss-Newton solvers. The performance is evaluated on synthetic navigation and real-world object-pushing data sets.

**Issues:**

See above.

**Reviewer Expertise:**

Fair: Some knowledge of the area

**Strengths And Weaknesses:**

Strengths
- The paper is well written and a pleasure to read.
- The method seems sound, and computationally efficient enough to be useful.
- The experimental results are encouraging.

Weaknesses:
I think it's a nice paper, but I have three significant concerns:
1) As a reason for ruling out "Loss 1", the authors claim that the mapping from \theta to \hat x in (3) is not differentiable. I don't understand this statement. Equation (3) states that \hat x is the minimizer of a smooth nonlinear least-squares problem. As such, it satisfies the first-order necessary conditions, and from this equation a simple application of the implicit function theorem gives the Jacobian of \hat x with respect to \theta. I think Loss 3 can still be justified on other grounds, but still this is strange.

2) Section 4.1 and 4.2 introduce the main ideas leading to Equation (7) and the associated Algorithm 1, and they read as though this is a novelty of the paper. I'm not an expert in this field, but I thought this was all completely standard in maximum-likelihood modelling. E.g. essentially the same approach is described in detail in Ch 18 of the Goodfellow et al Deep Learning book, under names such as Contrastive Divergence, e.g. Algorithm 18.1 in that book.

As far as I can see, the main novelty is using the estimated mean and covariance from the smoothing algorithm to generate the samples. The standard approach would be to use some sort of Monte-Carlo method (e.g. Gibbs sampling) to generate samples.

I think there is a nice contribution here, but this material could more clearly describe what is actually new in the approach and what is standard.

3) Related to the above, I like the experimental results, but the choice of comparisons is a bit strange. To me, a more natural comparison would be a standard approach to Loss 3 based on MCMC. E.g. you could still use the proposed method as a proposal distribution, but then run a few iterations of a Gibbs sampler and see if that makes a difference.



**Summary Of Recommendation:**

Overall, I found this an enjoyable paper to read and I think there are some useful ideas here. I have a few issues with the paper, as described above, but these should be possible to address in a revised version.

---

> ### Author Response · Authors · 2021-08-26
> **Response to Reviewer bQ4N**
>
> We thank the reviewer for their thoughtful and insightful comments and suggestions. We are really glad that the reviewer found the paper a pleasure to read. We discuss the major recommendations below:
>
> **Loss 1 Implicit function theorem**
>
> Loss 1 as we define requires differentiating through the optimizer process that is fundamentally a min operator and hence not differentiable. However, the reviewer is absolutely correct in suggesting that if one were to look at the first-order necessary condition, it says that the gradient of the cost evaluated at the ground truth must be zero [1]. We would place this particular loss as a separate category, e.g. Loss 4. Note that Loss 1 and Loss 4 are not equivalent for non-convex functions. Even if we found a \theta that drives Loss 4 to 0, upon using this \theta at test time, the optimizer may find itself in a completely different local minima from the ground truth.  Such problems don't necessarily reach a fixed point, thus a direct application of implicit function theorem is challenging. While this would be really interesting to explore, it is outside the scope of our current paper. However, we have updated the related work in our paper to call this option out explicitly.
>
> **Sections 4.1, 4.2 and Novelty**
>
> The reviewer is correct in pointing out that energy-based models are a broad framework used across many unsupervised learning applications. However, to the best of our knowledge, this is the first paper to make a connection between learning observation models end-to-end in state estimation / SLAM and energy-based learning. We use Sections 4.1 and 4.2 to instantiate the broad framework of energy-based learning within the context of our factor graph estimation problem.
>
> Indeed 4.3 makes the subtle, but important point that incremental Gauss-Newton graph solvers like iSAM2 [2] enable energy-based methods to scale to very large graphs. The key insight is to leverage sparsity and incrementally construct the Jacobian as new measurements arrive thereby enabling real-time construction of the Gaussian approximation.
>
> We have updated section 4 to better clarify these points.
>
> **MCMC comparison**
>
> Our focus with the current paper is on comparing LEO against alternate approaches to solve the end-to-end learning problem such as black-box search (Loss 1), LSTMs (Loss 1), and supervised learning on a surrogate loss (Loss 2). Doing an exhaustive comparison of various samplers with LEO is an important next step and a topic of future work.
>
> One thing to keep in mind with MCMC methods is that they generally do not scale well to high-dimensional problems. Our initial experimentation with an HMC sampler [3] seems to suggest very high runtimes on the 2D navigation datasets which have a 900-dimensional state space (3DOF * 300 steps). Hence for large factor graphs, MCMC may prove to be a major bottleneck. We will include an initial analysis of these runtimes as Appendix D in the final version. On the other hand, methods that exploit the sparsity of the graph, such as nonparametric belief propagation / mm-iSAM2 [4] could be better. However, this would indeed be out of the current scope.
>
> References
>
> [1]  B. Amos, and Z. Kolter. "Optnet: Differentiable optimization as a layer in neural networks." In International Conference on Machine Learning, pp. 136-145. PMLR, 2017.
>
> [2] M. Kaess, H. Johannsson, R. Roberts, V. Ila, J. Leonard, and F. Dellaert. iSAM2: Incremental smoothing and mapping using the Bayes tree. Intl. J. of Robotics Research, 31(2):216–235, Feb. 2012.
>
> [3] https://github.com/AdamCobb/hamiltorch
>
> [4] D. Fourie, J. Leonard, and M. Kaess. A nonparametric belief solution to the Bayes tree. In 2016 IEEE/RSJ International Conference on Intelligent Robots and Systems (IROS), 2016.

---

> > ### Author Response · Authors · 2021-08-31
> > **Follow-up on MCMC comparison**
> >
> > We have updated Appendix D to include an initial comparison to an MCMC sampler. Thank you for the suggestion!

---

### Official Review · Reviewer_krZP · 2021-07-27

**Originality:** Very Good
**Technical Quality:** Very Good
**Clarity Of Presentation:** Very Good
**Impact:** 4

**Recommendation:**

Strong Accept: I recommend accepting the paper and will argue for my recommendation even if other reviewers hold a different opinion.

**Summary:**

The paper proposes an algorithm for performing maximum-likelihood optimization of an energy-based model over states and observations that relies on Laplace + Gauss-Newton approximation to generate contrastive samples. The algorithm is evaluated by its tracking performance and sample efficiency on synthetic navigation and real-world planar pushing tasks.

**Issues:**

See weaknesses above

**Reviewer Expertise:**

Very good: Comprehensive knowledge of the area

**Strengths And Weaknesses:**

Strengths: The paper is well-written, clear, easy to follow. The approach is straightforward, interesting, and described in sufficient detail. The experiments are informative and highlight the properties of the algorithm. The video is well-done and provides a good overview of the approach.

Weaknesses: Section 3. Problem Formulation could almost entirely be removed. One could directly start with Eq. (6) that gives the standard ML objective for energy-based learning.

The Gauss-Newton method for inference is a well-known technique for sampling from intractable models. In the related work section and in Sec. 4.3 this should be clarified, because otherwise one gets an impression that the authors are claiming that they invented it. A recent overview of Laplace approximation with Gauss-Newton method, including links to prior work, can be found in [1].

The title seems misleading. In general, references to "graph" in the paper tend to obfuscate matters. In the end, what is meant is a "graphical model" or a "factor graph". But this is more-or-less standard in any paper on probabilistic modeling. One could therefore misinterpret "graph" in the title as though it were referring to graph neural networks or some algorithms for optimization over graphs. Perhaps the easiest fix would be to say "factor graph" in the title.

[1] Immer, A., Korzepa, M., & Bauer, M. (2021, March). Improving predictions of Bayesian neural nets via local linearization. In International Conference on Artificial Intelligence and Statistics (pp. 703-711). PMLR.

**Summary Of Recommendation:**

The paper considers an interesting idea of framing the observation model as an energy-based model. An approach for training this model is proposed and evaluated. The results appear encouraging and there are clear directions for future work provided. This paper may inspire further investigation into energy-based models for filtering and other applications in estimation and control.

---

> ### Author Response · Authors · 2021-08-26
> **Response to Reviewer krZP**
>
> We thank the reviewer for taking the time to provide a detailed and encouraging review. We greatly appreciate the feedback on the paper and the video. We agree that energy-based models are powerful and believe there is potential for making many more interesting connections to estimation and control. We go over the main recommendations below:
>
> **Section on Problem Formulation**
>
> While Section 3 may be straightforward for the ML community, it may serve as a useful entry point for the SLAM community who may not be as familiar with concepts such as energy-based learning. The prevailing consensus in the community is that the only way to train factor graph models is via unrolling. This problem formulation illustrates that it is far more natural to view factor graphs through the lens of energy-based methods. Hence, we are still inclined to leave that section in.
>
> **Gauss-Newton reference**
>
> We thank the reviewer for the reference and have updated our paper to include and discuss it. While we absolutely agree that the Gauss-Newton method is a well-known technique for sampling, we highlight perhaps the lesser-known connection between incremental Gauss-Newton approaches and energy-based methods. Incremental solvers like iSAM2 leverage sparsity and incrementally construct the Jacobian as new measurements arrive, thus allowing one to efficiently sample in real-time from very large graphs. We view this as an exciting connection between powerful tools from two different communities.
>
> **Title**
>
> We agree with the reviewer that the title can indeed be misinterpreted and have updated it to read as "LEO: Learning Energy-based Models in Factor Graph Optimization".

---

### Meta-Review · Area_Chair_ZpVz · 2021-08-20

**Recommendation:** Accept (Poster)
**Confidence:** 4

**Metareview:**

This paper proposed an algorithm for performing maximum-likelihood optimization of an energy-based model. The approach appears to be standard by applying Laplace and Gauss-Newton approximations. The authors should justify what is new. Additionally, the authors are expected to elaborate further and expand their comparison benchmark.

---

> ### Author Response · Authors · 2021-08-26
> **Response to meta-reviewer and all reviewers**
>
> We thank all reviewers and the meta-reviewer for their thoughtful feedback and comments. We appreciate that the reviewers find our approach impactful (krZP, bQ4N), with extensive experimental results (krZP, bQ4N, uKGu) and that it may inspire further investigation into energy-based models and other applications in estimation and control (krZP). We also appreciate that all the reviewers found our paper to be well-written, clear, and easy to follow (krZP, bQ4N, uKGu). We address the various points raised by the reviewers in our response and have updated our paper draft to reflect their feedback.
>
> We also capture below our arguments on novelty since it is an important concern. To the best of our knowledge, this is the first paper to make a connection between learning observation models end-to-end in state estimation / SLAM and energy-based learning. While factor-graph frameworks are widely used in SLAM and state estimation [1,2], there is very little work in the literature for learning observation or sensor models in factor graphs end-to-end -- typically models are hand-designed or learned independently.
>
> The only such work learning models end-to-end require factor graph optimizer to be differentiable and do this via unrolling [3,4]. However, most real-time SOTA factor graph libraries are not differentiable. Moreover, our initial analysis in Appendix A shows that differentiable unrolling approaches can easily overfit on simple 1D problems and fail to recover the correct model.  On the other hand, since we only learn the shape of the energy function which is independent of the optimizer, our approach is robust to the optimization procedure.
>
> In the context of energy-based methods, our novelty lies in the specific way we build a Laplace approximation using a popular “incremental” Gauss-Newton solver iSAM2 [5] in SLAM that can scale to sampling efficiently from very large graphs. The key insight is to leverage sparsity and incrementally construct the Jacobian as new measurements arrive thereby enabling real-time solutions.
>
> Finally, in the context of real-world tactile planar pushing application, this is the first paper to argue and show that observation models can be learned end-to-end. The previous SOTA method [6] learned observation models independently and showed that it outperformed hand-designed baselines. We go a step further and show evaluations where learning observation models end-to-end outperforms this approach.
>
> We view our work as bridging powerful techniques from two distinct communities -- real-time graph inference from the SLAM community and energy-based learning from the machine learning community. We believe this will inspire further collaboration between the communities.
>
> References:
>
> [1] F. Dellaert and M. Kaess. Factor graphs for robot perception. Foundations and Trends in Robotics, 6(1-2):1–139, 2017.
>
> [2]​​ C. Cadena, L. Carlone, H. Carrillo, Y. Latif, D. Scaramuzza, J. Neira, I. Reid, and J. J. Leonard.  Past, present, and future of simultaneous localization and mapping:  Toward the robust-perception age. IEEE Trans. Robotics, 32(6):1309–1332, 2016.
>
> [3] B. Yi, M. Lee, A. Kloss, R. Martin-Martin, and J. Bohg. Differentiable factor graph optimization for learning smoothers. arXiv preprint arXiv:2105.08257, 2021.
>
> [4] M. Bhardwaj, B. Boots, and M. Mukadam.  Differentiable Gaussian process motion planning.  In IEEE Intl. Conf. on Robotics and Automation (ICRA), 2020.
>
> [5] M. Kaess, H. Johannsson, R. Roberts, V. Ila, J. Leonard, and F. Dellaert. iSAM2: Incremental smoothing and mapping using the Bayes tree. Intl. J. of Robotics Research, 31(2):216–235, Feb. 2012.
>
> [6] P.  Sodhi,  M.  Kaess,  M.  Mukadam,  and  S.  Anderson.   Learning tactile models for factor graph-based estimation. In IEEE Intl. Conf. on Robotics and Automation (ICRA), 2021

---

### Decision · Program_Chairs · 2021-09-13

**Decision:**

Accept (Poster)

**Comment:**

This paper proposed an algorithm for performing maximum-likelihood optimization of an energy-based model. The approach appears to be standard by applying Laplace and Gauss-Newton approximations. The authors should justify what is new. Additionally, the authors are expected to elaborate further and expand their comparison benchmark.